# The Role of the Specific Strength Test in Handball Performance: Exploring Differences across Competitive Levels and Age Groups

**DOI:** 10.3390/s23115178

**Published:** 2023-05-29

**Authors:** Luis J. Chirosa-Ríos, Ignacio J. Chirosa-Ríos, Isidoro Martínez-Marín, Yolanda Román-Montoya, José Fernando Vera-Vera

**Affiliations:** 1Physical Education and Sports Department, Faculty of Sport Sciences, University of Granada, 18011 Granada, Spain; lchirosa@ugr.es; 2Sports Science Department, Faculty of Sport Sciences, University of Leon, 24004 Leon, Spain; imarm@unileon.es; 3Statistics and Operational Research Department, Faculty of Sciences, University of Granada, 18071 Granada, Spain; yroman@ugr.es (Y.R.-M.); jfvera@ugr.es (J.F.V.-V.)

**Keywords:** team handball, specific physical tests, players selection, field tests

## Abstract

The aim of this study was to determine if specific physical tests are sufficiently discriminant to differentiate players of similar anthropometric characteristics, but of different playing levels. Physical tests were conducted analyzing specific strength, throwing velocity, and running speed tests. Thirty-six male junior handball players (*n* = 36; age 19.7 ± 1.8 years; 185.6 ± 6.9 cm; 83.1 ± 10.3 kg; 10.6 ± 3.2 years of experience) from two different levels of competition participated in the study: NT = 18 were world top-level elite players, belonging to the Spanish junior men’s national team (National Team = NT) and A = 18 players of the same age and anthropometric conditions, who were selected from Spanish third league men’s teams (Amateur = A). The results showed significant differences (*p* < 0.05) between the two groups in all physical tests, except for two-step-test velocity and shoulder internal rotation. We conclude that a battery combining the Specific Performance Test and the Force Development Standing Test is useful in identifying talent and differentiating between elite and sub-elite players. The current findings suggest that running speed tests and throwing tests are essential in selecting players, regardless of age, sex, or type of competition. The results shed light on the factors that differentiate players of different levels and can help coaches in selecting players.

## 1. Introduction

Handball is a high-intensity team sport, where explosive actions with high strength involvement, such as jumping, sprinting, throwing, and changing direction, are fundamental for sport performance [1,2,3,4,5,6]. Players must perform frequent periods of high-intensity activity with highly variable recoveries over time, combined with moments of low intensity [4,7,8]. Further, for top-level players, decision making, as well as other conditional and biometric factors will determine the level of the player (reference). The first ones could be defined as internal factors (cognitive) and the second ones as external factors (conditional and biometric).

Although handball is a team sport, as any other sport, where the causes of performance are multifactorial, strength is one of the most determining external factors due to the high involvement it has in fundamental gestures such as throwing, jumping, changes in direction or impacts-contacts [9,10,11,12]. When assessing strength, it is necessary to apply tests capable of both discriminating analytical improvements in athlete strength production, as well as its transfer and application to the game. The way to be able to evaluate it is a matter of concern for researchers of Sport Sciences, especially when they want to use it to classify and discriminate athletes of different levels, categories, sex, etc. A good evaluation protocol should differentiate a good player from a not so good player, in addition to identifying the rates of improvement during a season.

It can be complicated to choose the type of test intended to define performance and classify players for selection in a team or national team [13,14]. Among the most commonly used factors in handball evaluation is anthropometry [15] along with strength assessment tests [16] and field application tests [17]. In the literature of the last decade, several evaluation proposals have appeared, more or less specific, combining assessment of anthropometric characteristics together with their capacity for force production and application in handball players of different levels, age and sex [18,19,20,21,22]. Previous work has shown that there are differences between elite and amateur handball players in terms of strength manifestations, such as maximal dynamic strength and muscle power development, as well as anthropometric characteristics [2,23]. Several studies have observed how anthropometric differences are determinants in the performance of handball players [15,21,24,25,26], favoring the application of explosive strength to determinant gestures in the game such as throwing [27]; in fact, it seems clear that greater anthropometry is clearly related to an increase in game performance and that it is a differential factor in any category [18,25,28].

Although the evidence for performance between anthropometry (external factor) and level of play is clear, there is an underlying question little studied in our sport that is related exclusively to the level of play and the production of force, what happens when players of different levels, but of the same age have the same anthropometric characteristics? Are there still differences in their physical ability to manifest and apply force? Recent studies in other sports, such as soccer, have partially answered this question [29] and these works have shown that players of different competitive levels with similar anthropometric conditions have different physical performances, concluding that elite soccer players show better performance indicators in the strength variables studied [29].

Studies on male handball players have shown that both physiological and physical characteristics differentiate players according to category, sex and level [26,28,30,31,32,33]. The problem is that in all these investigations, the factors of age, sex and anthropometry are not kept fixed in order to analyze exclusively the physical factors. Therefore, it would be of great interest to conduct a comparison between players with different training experiences, keeping age and anthropometry fixed so that they cannot act as contaminating variables. It would also be important to know whether a given battery of tests, in which the application of strength is an essential component, is able to discriminate the level of the players by itself without other factors being present and thus avoid test redundancies.

Consequently, the present study aims to examine the differences in physical performance between non-elite and elite junior handball players, keeping anthropometry and age stable. The aim is to determine whether the force production capacity and its application in field tests are sufficient to differentiate or classify selected junior players at the highest level from their non-elite peers, regardless of their anthropometry. At the same time, the aim is to check which test can be more determinant to differentiate players who should be eligible for selection from those who are not.

## 2. Materials and Methods

Thirty-six male junior handball players (*n* = 36; age 19.7 ± 1.8 years; 185.6 ± 6.9 cm; 83.1 ± 10.3 kg; 10.6 ± 3.2 years of experience) from two different levels of competition participated in this study: National Team = 18 were world top-level elite players (*n* = 18; age 20.2 ± 0.8 years; 185.6 ± 7.5 cm; 83.1 ± 10.5 kg), belonging to the Spanish junior men’s national team (National Team = NT) and A = 18 players of the same age and anthropometric conditions (*n* = 18; age 19.7 ± 1.0 years; 185.6 ± 4.5 cm; 83.1 ± 10.2 kg), who were selected from Spanish third league men’s teams that had not been selected as amateur players (Amateur = A). This study adopts a Quasi-Experimental Design to compare and assess differences between the groups under investigation.

The measurements were taken before the start of the competition season; all the tests were performed on the handball court. This study was approved by the Ethics Committee of the University of Granada, Spain, and was conducted in accordance with the requirements established in the Declaration of Helsinki (2/2020). The participants were informed about the procedures to be performed and written informed consent was obtained from all of them. Information on age, training experience, supplementation and presence of injuries was collected by means of a personalized interview. All the players had undergone previous medical tests; only those athletes who were not injured at the time of the test participated in this study.

To reduce the influence of uncontrolled variables, all participants were instructed to maintain their typical lifestyle and dietary habits before and during the study. Subjects were told not to exercise the day before the test and to consume their last meal (without caffeine) at least 3 h before the scheduled test. In addition, they drank at least 0.5 L of pure water during the last hour before the test. They were also asked to sleep regularly before the protocol. During all performance-based tests, athletes were instructed to perform at their maximum capacity.

### 2.1. Procedures

The subjects were familiar with the testing protocol, for which all tests had passed a reliability test (Hopkins, 2000) prior to the familiarization period, with the ICC being above 0.9 and CV less than 10%, thus it was considered that the tests were learned by the players.

The physical test battery consists of two different blocks:

The Specific Performance Test: Two tests of strength application related to the game—for the upper body, the throwing velocity that deals with throws at maximum speed with three steps (Chirosa-Ríos et al., 2020); and for the lower body, the 30 m sprint test (30 mST) was used, which is a maximum speed run of 30 m taking the time in 10–20–30 m.

The Functional Dynamometric Strength Test: Two tests that measure the manifestation of strength in a more specific way through the use of a functional electromechanical dynamometer (DEMF) [34]. An isometric test for the lower body is the shoulder rotation test [35] and the two-step test (TST) [36], which is a dynamic test that evaluates the maximal manifestation of strength in the lower body. In our case, we have chosen, from the multitude of data, the speed of displacement (two-step test—velocity, TSTv) and the power of the performance of the gesture (two-step test—force, TSTf).

#### 2.1.1. Anthropometric Parameters

Prior to testing, participants were assessed for height and body mass. Height was assessed to the nearest 0.001 m, using a stadiometer (Holtain Ltd., Crymych, UK). Body mass and body fat percentage were measured to the nearest 0.1 kg, using an electronic scale (Seca Instruments Ltd., Hamburg, Germany). To determine the participants’ body mass index (BMI), the measured values were used in a standard calculation.

#### 2.1.2. Handball Test Protocol

Prior to the performance of the test battery, the players were given verbal instructions and shown videos about the different tests in which they were to participate. Each participant was carefully instructed and verbally encouraged to give their maximum performance in the tests. Each test was supervised by a handball specialist, and only those that complied were recorded.

The warm-up was standardized, consisting of soft running, static and dynamic range of motion, 5 progressive speed runs of 30 m, 2 series of 10 push-ups and 2 series of 6 throws progressively looking for maximum speed. The test battery was divided into two test blocks: the Specific Performance Test and the Functional Dynamometric Strength Test. Each of these parts had two tests in order not to fatigue the players and not to intervene too much in the preparation of the teams, the order of application was randomized.

#### 2.1.3. Throwing Velocity

The participants performed the running throw as executed in the game, behind the throwing line at a distance of 9 m.

Throws were performed with an official ball according to IHF/EHF regulations (ball weight 425 g, −475 g, ball radius = 58–60 cm), and subjects were allowed to use resin according to convenience.

Five throws were measured—of the five throws, the last four were scored for this study. Each throw was measured in kilometers per hour (km/h) using a radar (Stalker ATR, Professional radar, Applied Concepts Inc., Plano, TX, USA), with an accuracy of ±0.1 km/h and a sampling frequency of 100 HZ within a field of action of 10° where the pistol was placed behind the goal on the thrower’s axis at the height of his arm. The participants were placed in the corresponding position and performed the throw, with a rest between throws of 1 min. High test–retest reliability was found (ICC = 0.97, CV = 4%).

#### 2.1.4. The 30 m Sprint Test

Each player was asked to perform a sprint as fast as possible standing with both feet shoulder-width apart, 5 cm behind the first timing gate. To measure sprint time, 3 light beams (Brower Timing System CM L5, Brower, UT, USA) placed at 10, 20 and 30 m from the test distance were used. Each subject had to repeat the sprint test twice, with 2 min recovery between tests. The fastest sprint time of 30 m was used for the calculation. High test–retest reliability was found (ICC = 0.93, CV = 2%).

The Functional Dynamometric Strength Test:

#### 2.1.5. Standing Shoulder Internal Rotation

Subjects were positioned standing and supporting the dominant upper limb on a subjection system of own manufacture. The subjection system was regulated, taking into account the subject’s height with a variation of ±1 cm. The humerus was fixed with a cinch at 2/3 of the distance between the lateral epicondyle and the acromion. Position was determined with a baseline goniometer (Gymna hoofdzetel, Bilzen, Belgium). The position consisted of a 90° adduction of the glenohumeral joint and a 90° flexion of the humeroulnar joint. For the glenohumeral joint, the fulcrum was positioned in the acromion with the vertical arm stable and the arm movable along the humerus with the lateral epicondyle as a point of reference. For the humeroulnar articulation, the fulcrum was positioned in the lateral epicondyle with the arm stable in horizontal and the arm movable along the forearm with the processus styloideus ulnae as a reference point.

Participants first attend (four subjects each time) in a well-rested condition at the start of each testing session of 45 min with the FEMD. The protocol consisted of a general warm-up for both test session consisted on 5 min of jogging (beats per minute < 130; measurement with a Polar M400), 5 min of joint mobility and 2 sets of 6 s of internal rotation and external rotation in the previous stablished position. After the warm-up of familiarization protocol, participants rested for 5 min before the initiation. The test consisted of two series of 6 s of shoulder internal and external rotators. The rest between sets was a three-minute. The mean and peak force were taken to calculate the mean dynamic force for each participant, (ICC = 0.97, CV = 3%).

#### 2.1.6. The Two-Step Test

Participants stood with their feet shoulder-width apart. They had to perform a two-step forward movement, touching at the end a person simulating an opponent. The final position was standing with the second leg they moved forward and holding an “opponent”. Any initial or final position was previously established based on one’s own freedom of movement. An appropriate belt was used to avoid injury when performing the explosive forward steps. A free range of motion was established without taking any measurements.

The test consisted of two sets of six consecutive maximum repetitions, with 15% body weight overload with free range of motion. Participants were required to perform a gesture similar to the forward movement of holding an opponent in two steps in handball at the maximum possible speed. There was a three-minute pause between sets. Only the maximum velocity, TSTv, and the power of each repetition, TSTf, were taken as variables.

Body displacement velocity was evaluated with a FEMD (Dynasystem Health, Symotech, Granada, Spain) with a precision of 3 mm for displacement, 100 g for a sensed load, and a range of velocities between 0.05 m·s^−1^ and 2.80 m·s^−1^, coupled with a standard bench, an appropriate hip belt, a pulley system and a subjection system (ICC = 0.96, CV = 4%).

## 3. Statistical Analysis

All data were analyzed considering the group to which each player belongs. Descriptive measures (mean, standard deviation and 95% confidence intervals) were calculated for the anthropometric variables, testing the difference between the mean values of each group using a general linear model. Differences between means were considered statistically significant if the *p*-values are less than 0.05 and the effect size (eta) takes values greater than 0.10.

For the performance variables, the Strength Transfer Field Test and the Specific Strength Test, a general linear model was used to analyze the differences between the two teams by looking at the results of both tests. The mean values obtained by each player after the series of repetitions of the different tests are considered. The normality of the observations was previously verified. The differences between the mean values of the results obtained were considered statistically significant if the *p*-values are less than 0.05 and the effect size (eta) takes values greater than 0.10. The confidence intervals (at 95% confidence) of the mean values were also included.

Linear discriminant analysis was used to determine the most influential variables in classifying a player as belonging or not to the national team, as well as to determine a linear discriminant rule to determine the classification of any new player to one of the two groups. Taking into account the sample size, the hypotheses of the model were checked for each variable by means of Shapiro–Wilks and Levené contrasts. The hypotheses of equality of means and multivariate homoscedasticity were tested with Wilks’ Lambda test and Box’s test. The most determinant variables were selected using Wilks’ method and Mahalanobis’ distance, and the ranking results for the players in the sample were also analyzed with a cross-validation procedure to determine the ranking results. All analyses were performed using SPSS Statistics 27 (IBM Corp., Armonk, NY, USA).

### 3.1. Comparative Analysis

Analyzing the anthropometric characteristics of the players according to the team to which they belong, it is observed that there are no significant differences between the main variables studied, considering the confidence intervals and *p*-value associated with the contrast of equality of mean values between groups (Table 1).

In view of the results, regarding the physical tests performed, there were significant differences in the two groups of tests of the battery used, the specific performance test (SPT) and the Functional Dynamometric Strength Test (FDST) (Figure 1).

In the SPT, the displacement velocity (m/s) in the first section 10 m speed was 1.49 ± 0.06 NT, versus 1.74 ± 0.07 A (*p* < *0*.001; ES = 3.43); in the second section of 20 m speed it was 2. 71 ± 0.10 NT, versus 3.01 ± 0.10 A (*p* < 0.001; ES = 2.56); in the third stretch of 30 m speed it was 3.86 ± 0.15 NT, versus 4.11 ± 0.34 A (*p* < 0.008; ES = 0.93). In ball displacement speed (KM/h) the results were 102.44 ± 5.49 NT, versus 81.00 ± 4.95 A (*p* < 0.001; ES= −4.09).

In the FDST, the two-step test revealed a significant difference in power (W) between the force results: 1128.72 ± 172.7 NT versus 817.00 ± 141.87 A (*p* < 0.001; ES = −1.97).The two-step test velocity results were 2.65 ± 0.06 NT, versus 2.65 ± 0.01 A (*p* < 0.001; ES = −1.01 A), 2.65 ± 0.01 A (*p* = 0.984; ES = 0.01) and in the shoulder internal rotation the results were 22.98 ± 4.9 NT, versus 22.26 ± 4.47 A (*p* = 0.0621; ES = −0.17). 

### 3.2. Discriminant Analysis

The influence of the variables, BMI, body height, body weight, body fat, body mass, 10 m speed, 20 m speed, 30 m speed, throwing velocity, two-step test—force, two-step test—velocity, shoulder internal rotation to determine the membership of a player to the selection has been analyzed using linear discriminant analysis with SPSS. The variables were first standardized by their range to avoid measurement differences.

The usual assumption for the linear discriminant analysis is performed. First, the normality and homoscedasticity of each variable among the two groups of players (1 = selection, 2 = national club) were analyzed. The Shapiro–Wilks normality test was significant at the level 0.05 for the variables BMI (*p* = 0.027), body fat (*p* = 0.005) and 30 m speed (*p* = 0.004), both for the group of the national selection players, while the Levené’s test indicates lack of homoscedasticity for total water (*p* = 0.002) and shoulder internal rotation med (*p* = 0.026). Hence, the influence of variables body mass, 10 m speed, 20 m speed, throwing velocity, two-step test—force and shoulder internal rotation max was analyzed to determine a discriminant rule for the membership of a player to the national selection. Significant differences are identified for each variable in terms of the Wilks’ Lambda test of equality of groups means, except for body mass (*p* = 0.257), two-step test—force (*p* = 1) and shoulder internal rotation max (*p* = 0.621). Second, the Box’s test of equality of covariance matrices showed a *p*-value of 0.915 which does not contradict the multivariate homoscedasticity hypothesis, and a Wilks’ Lambda value of 0.088 indicates that almost all the variance is explained by group differences, which related to a *p*-value of 0, indicates highly significant differences between the two group centroids (Chi-square value of 74.047).

A stepwise procedure was used to determine the influent variables in the linear discriminant procedure. The variables related to the small value of the Wilks’ Lambda, which in this data sets also coincides which that maximizing the Mahalanobis distance between the two closets groups, were throwing velocity, 10 m speed and 20 m speed, for Wilks’ Lambda values of 0.202, 0.146 and 0.149, respectively, while the remaining variables were not included in the analysis. The standardized canonical discriminant function coefficient were −1.622 for 10 m speed, −1.119 for 10 m 20 m speed and 0.750 for throwing velocity, which shows the SPRINT10 and SPRINT20 variables appear to have the greatest impacts.

Considering the above results, linear discriminant analysis is performed for the resulting variables. For each group, the discrimination functions to classifying a player in the group related to the highest score (NT = National Team or A = club amateur) are:NT = −32.770 × 10 m speed + 208.50 × 20 m speed + 127.212 × throwing velocity −490.280
A = −102.380 × 10 m speed + 251.599 × 20 m speed + 158.714 × throwing velocity −513.625

For the analyzed sample, the overall success of the three variables in the model for classifying cases into one of the two groups is 100%. Since the sample is small, cross-validated classification results (for one leave-out player) also showed a 100% of correct results.

## 4. Discussion

The main objective of this study was to determine whether the force production and application capacity of junior elite handball players were different from that of their non-selected peers, independently of their anthropometry, one of the most studied factors together. At the same time, we wanted to know if the physical tests are sufficiently discriminant to differentiate players of similar anthropometric characteristics, but of different playing levels. The results have clearly shown that there are significant differences *p* < 0.05 between the two groups in all the physical tests analyzed, both for the lower and upper body, except in the two-step test velocity and the shoulder internal rotation.

To the best of our knowledge, this study represents the first analysis of strength capabilities among handball players of varying levels, while controlling for age and anthropometric factors. We have sought to know the differences in two types of tests: on the one hand the SPT, close to the specific skills related to performance and the player’s ability to apply force, for this purpose we chose the throwing velocity and the 30 m speed which are the most used by coaches and researchers and on the other hand we applied the FDST, with the standing shoulder internal rotation and the two-step test for the upper and lower body, respectively, these tests were chosen because they are the most related to the gestures of competition.

After the analysis of the results it is clear that a battery combining the SPT and the FDST, is useful to facilitate the discrimination of elite players from sub-elite players, which gives a basis for identifying talent [18,19,28,37]. It was known that anthropometry is a factor that plays in favor of performance in handball players in important gestures such as throwing and displacement and that it is also a differentiating factor between levels of play [19,26,27]. However, to date, it was unknown what happened in force production capacity and its application to gestures close to those of competition in players of similar anthropometric characteristics and age. The selected variables enable discrimination between an elite group and another. Based on these variables, it is more likely to predict that an individual is part of the elite group when they have high values in stronger, faster, throw, and speed. Furthermore, these differences could be attributed to the more qualitative and targeted training undertaken by high-level athletes, although the specific quantification of these values was not conducted in this study, which would be an area of significant interest for future research. The discriminant analysis confirms these findings as well. Our research is in line with recent work carried out in soccer, where it has been shown that players of different competitive levels (elite and non-elite), with similar anthropometric conditions had very different physical performances [29].

Partially analyzing the results, as the most outstanding data, indicates that in the SPT, the two tests applied to the selected players were significantly superior to the amateur group (*p* < 0.001), over 20% faster in the speed tests and 21% in the throwing tests. These results are in line with other investigations with similar objectives to ours, [28] for example, found large differences between categories of play in the German handball league (professional and amateur level) in the assessment of throwing speed, concluding that strength, power and throwing speed are important and discriminating factors in professional handball. Similar results have been given in other studies related either to throwing ability or to speed, but differences in the type of sample or in the test battery applied make direct comparison difficult [11,15,19,26]. 

It can be suggested after the analysis of the related literature and with the results of our study that in the selection of players, regardless of age, sex and type of competition travel speed tests and throwing tests, such as the ones performed here, are tests that allow differentiating levels of play and therefore should be used by coaches to know their players [11,15,19,28]. The reasons for these differences may be diverse, as they may be related to intra- and intermuscular synchronization, to greater motor coordination on the part of the players, to maturity itself, etc., which clearly should be a reason to study for new research to shed light in this field. 

On the other hand, analyzing the effects of the tests related to the FDST, it should be noted that, in this case, only the power in the two-step test shows significant differences between groups. This is a test clearly related to the measurement of force production capacity in a basic handball gesture that is used both in attack and defense and that allows direct measurement of specific displacement power thanks to the use of the DEMF. What is relevant in our results is that the elite players almost doubled their amateur peers (68% difference), maintaining practically the same execution velocities (two-step test—velocity = 2.65 m/s). Since it is a free gesture similar to the one used in the game in an acceleration of unmarking or in a defensive action, if more force is produced at the same velocity, it is clearly indicating that the player can apply more power. To our knowledge, this is the first study where the DEMF has been applied to FDSTs, which makes it very difficult to compare with other works performed in handball, since these usually use generic tests, such as bench press or squats and other technological measurement devices [38,39]. Regardless of the type of test, if we only take into consideration force production, the results coincide with other investigations that see large differences between 20 and 40% between elite and amateurs in the ability to manifest force [19,31,38,40].

The second objective of this research was to test which tests could be more determinant to differentiate selectable players from non-selectable players. This analysis, in addition to being novel for junior players, is important because it allows coaches, if necessary, to reduce the number of tests to be applied or to prioritize the result of some tests over others when establishing a test battery. Especially in junior players, as in this case, that although they are still in a training process, their level of biological maturation is high and therefore anthropometric factors will not undergo major changes or be affected by age and maturity as happens in other lower categories [41,42].

After the analysis, being very cautious due to the size of the sample, we can say that from the battery of tests applied, the tests that clearly discriminate the players are the SPT, being the 10 m speed the most discriminating, followed by the 20 m speed and the throwing velocity (the coefficient of the standardized canonical discriminant function was −1.622 for 10 m speed, −1.119 for 20 m speed and 0.750 for throwing velocity). It should be noted that the overall accuracy of the three variables of the two classification models used was 100%, having a small sample it was decided to refute the case model with the cross-validation model, as can be seen the results are convincing when it comes to discriminating the players. It is likely that the large differences in the level of the selected sample are what facilitate the discrimination.

Data on discriminant variables in handball players are scarce, so it is relatively difficult to compare the results—there is little research and the variables studied for discriminant analysis and the type of samples differ from our study [26,32,43,44]. In a similar study in which they included part of the variables studied here, but in younger players and therefore exposed to clearer maturational processes, they also concluded that throwing velocity had the strongest relationship with the discriminant function (Wilk’s λ = 0.502, χ^2^ = 57.21, *p* < 0.001); however, displacement did not come out as a discriminant factor in this function [22]—the difference with our work may be in the type of sample analyzed, higher biological maturation and the influence of sex.

Although as we have commented that it is not possible to make direct comparisons when discriminating performance factors that can influence the selection of players, due to the disparity of sample and procedures, in light of the literature reviewed, it is evident that throwing and/or pitching are discriminating factors between players of different levels or sex and can be used as evidence for this purpose and this is undoubtedly an important requirement to be among the elite [21,38,39,41,45,46]. It also appears that conditioning variables that have to do with the application of force, such as throwing and running, are more discriminant for boys than for girls [14,46].

To conclude, we agree with [18], who evidenced that most of the batteries and tests that are applied to collective sports players are conducted outside of real game situations, thus eliminating possible sources of very valid information to really achieve the objective of facilitating the discrimination of players. It would be of interest to further expand this line of research by incorporating control variables within the actual competition setting. The utilization of micro-technology sensors [4,5,6] already enables the possibility of employing discriminant analysis to select players based on factors such as displacement speed, accelerations, and throwing speed, which are crucial indicators of performance. These variables could effectively discriminate between different levels of play, even when keeping anthropometric values constant, as demonstrated in this particular case.. More studies along these lines are needed to facilitate and clarify these talent selection processes, using the fewest number of variables.

## 5. Conclusions

With this study, it is demonstrated, with a small sample, but of great sporting level, that not only is anthropometry the key to reaching elite levels, but other factors such as the ability to manifest strength in specific skills will play a factor, concluding that:- There are clear significant differences in the ability to produce and apply force in the throwing velocity, the 30 m speed and the two-step test between elite and amateur players of the same age and anthropometry.- The tests that discriminate junior elite handball players are the SPT, with the 10 m speed the most discriminating, followed by the 20 m speed and the throwing velocity.

## 6. Practical Applications

Our research has shown that the use of specific strength assessment tests, the FDST, together with field tests, the SPT, is useful for the trainer or researcher to know the operational status of the athlete at a given time without deviating too much from the specific preparation objective, since these tests are suitable to evaluate and improve performance at the same time. In this research, FDSTs could be applied thanks to the use of the DEMF, a new device that allows evaluating and training strength in natural conditions [34,47,48,49,50].

Based on these data and the studies analyzed, we can indicate that players with high speeds over short distances, no more than 20 m, and with power in the throw will have an advantage over the rest. Undoubtedly, these are going to be key factors to reach higher levels of performance in handball and therefore should be considered when choosing a player and designing training sessions. Our findings provide information for the assessment and evaluation of talents, allow differentiating players of the same anthropometry and age, and indicate that the batteries that combine strength production, especially with gestures similar to those of competition, and field tests are useful to facilitate the decision to choose players.

A more concrete contribution is that it is possible to discriminate the level of the elite handball player according to his maximum speed in the first 10 or 20 m as well as his throwing ability. These data will allow those in charge of selecting players in the final stages of maturation, such as juniors, to focus their attention on explosive tests as close as possible to the competitive game and to reduce, if necessary, the number of tests to be applied.

## 7. Limitations

The main limitation of this study is the size of the sample, and this limitation has been caused by the type of sample we have sought—elite players of the highest level compared with their not so successful peers. Therefore, these data should be interpreted with caution and in comparison with similar research.

Despite the inherent methodological limitations, the use of a cross-sectional comparative study, when using a top-level sample, can have important implications in the sporting arena, guiding coaches in their daily work and providing scientists with meaningful information to develop future research.

## Figures and Tables

**Figure 1 sensors-23-05178-f001:**
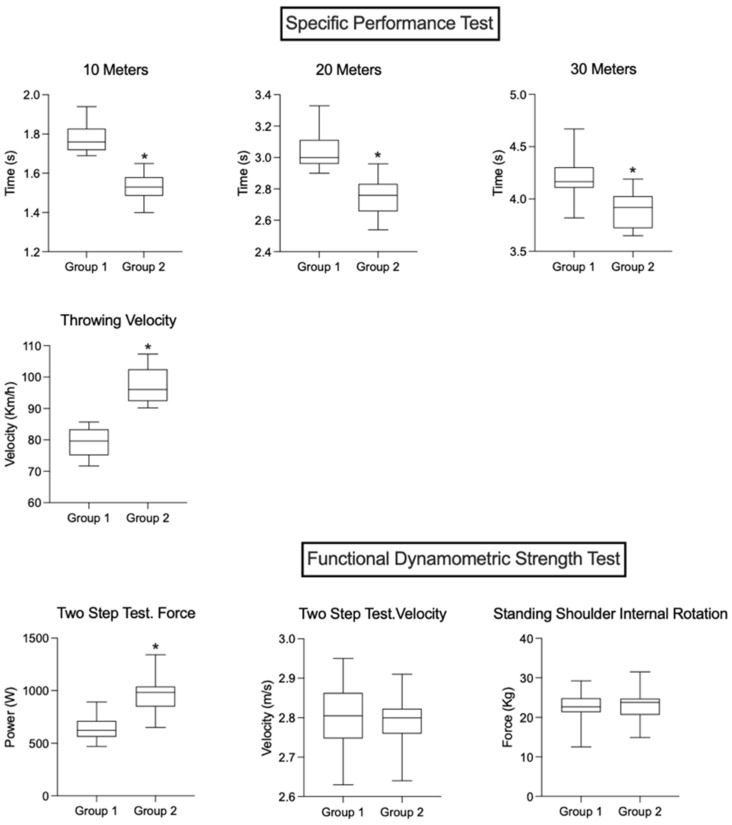
Physical fitness relationship of the National Team (*n* = 18) vs. the amateur player (*n* = 18). * An asterisk means statistically significant differences between groups (*p* < 0.001).

**Table 1 sensors-23-05178-t001:** Anthropometric characteristics of the National Team (*n* = 18) and amateur players (*n* = 18).

	National Team			Amateur				
Anthropometric Characteristics	Means	SD	SE	Means	SD	SE	P	ES
Age (yrs)	20.22	0.80	0.19	19.72	2.51	0.59	0.428	−0.268
Body Height (cm)	185.61	7.51	1.71	185.44	6.74	1.59	0.945	−0.023
Body Weigh (kg)	86.78	10.52	2.48	85.56	10.22	2.48	0.288	−0.360
Body Fat (%)	14.55	4.16	0.98	13.70	3.58	0.84	0.233	−0.324
Body Mass (Kg)	74.43	17.22	4.05	73.08	9.35	2.73	0.141	0.502
ES = Coen´s d								

## Data Availability

Data is unavailable due to privacy or ethical restrictions of the Spanish Handball Federation.

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
