# Peer review of "The Role of the Specific Strength Test in Handball Performance: Exploring Differences across Competitive Levels and Age Groups"

_sensors, 2023, doi:10.3390/s23115178_

Round 1

Reviewer 1 Report

Comments in the text

Author Response

Dear reviewers/colleges,

Thank you sincerely for your valuable time and dedication in reviewing our manuscript. We greatly appreciate your thorough assessment and constructive comments, which undoubtedly contribute to enhancing the quality of our work.

We have taken each comment into careful consideration and made the necessary revisions accordingly. To ensure transparency and ease of identification, we have highlighted all the changes made to the manuscript in yellow. This allows for a clear distinction between the original content and the modifications implemented in response to your insightful feedback.

Should you require any further clarification or have additional suggestions, please do not hesitate to let us know. We are committed to addressing any remaining concerns to the best of our abilities.

Thank you for your continued support and guidance throughout this review process.

Sincerely,

The authors

Comment: Page 1 line 22; I don't think that the word junior is suitable for this age group

Authors’ reply: Thank you very much for your comment. In the context of our study, the term "Junior" refers to the age category division as defined by the International Handball Federation. This categorization helps to organize and classify the different age groups in the sport of handball, ensuring fair competition and appropriate development opportunities for players.

Comment: Page 2 line 68; What do you mean with '' greater anthropometry''. Re-phrase please....

Authors’ reply: Thank you for your comment. We use that phrase to indicate that being taller and having larger extremities have consistently been demonstrated to have a positive impact on performance in handball. These physical attributes contribute to the facilitation of crucial actions, such as powerful throws. We have rephrased the sentence to enhance clarity and improve the reader's comprehension of the text.

Comment: Page 2 line 75; strength or force. Which are the difference for you? Please use one of them....

Authors’ reply:  Thank you very much for your comments. Force refers to the applied push or pull, while strength represents the muscular capacity to generate force. Force is a physical quantity, and strength is a physiological characteristic related to muscle performance. So, we don’t think that it has to be change in the manuscript.

Comment: Page 2 line 78; add references.

Authors’ reply: Dear reviewer, Thanks for your help, we have added the reference: Tereso, D.; Paulo, R.; Petrica, J.; Duarte-Mendes, P.; Gamonales, J.M.; Ibáñez, S.J. Assessment of Body Composition, Lower Limbs Power, and Anaerobic Power of Senior Soccer Players in Portugal: Differences According to the Competitive Level. Int. J. Environ. Res. Public Health 2021, 18, 8069. https://doi.org/10.3390/ ijerph18158069

Comment: Page 3 line 104; add ''National team''

Authors’ reply: Thanks for the comment, it has been changed accordingly.

Comment: Page 3 line 138; what do you mean with ''upper train'' & "lower train''?

Authors’ reply: Thank you for your comment. In the context of the mentioned phrase, we specifically refer to the body region where the test holds the greatest significance or implication. It was a mistake in the used term, we wanted to say “upper body” & “lower body” it has been changed accordingly.

Comment: Page 3 line 149; add reference for this test.

Authors’ reply:  Thanks for your help, we have added the reference: Chirosa-Ríos, I., Ruiz-Orellana, L., Jerez-Mayorga, D., Chirosa-Ríos, L., del-Cuerpo, I., Martínez-Martín, I., Rodríguez-Perea, Á., Pelayo-Tejo, I., & Martinez-Garcia, D. (2022). DEFENSIVE TWO-STEP TEST IN HANDBALL PLAYERS: RELIABILITY OF A NEW TEST FOR ASSESSING DISPLACEMENT VELOCITY [Prueba defensiva de dos pasos en jugadores de balonmano: Fiabilidad de una nueva prueba para evaluar la velocidad de desplazamiento]. E-Balonmano.com: Revista de Ciencias del Deporte, 18(3), 233-244. Recuperado de http://ojs.e-balonmano.com/index.php/revista/article/view/613

Comment: Page 4 line 150; in the results presented % of body fat. How was it measured?

Authors’ reply: Thank you for your comment. The values mentioned in our study were directly obtained from a Body Mass Device (bio-impedance), and these values were expressed as percentages.

Comment: Page 4 line 174; which test was performed first? was it the same for all athletes? can the order affect the results?

Authors’ reply:  Thank you for your comment. To address any potential doubts regarding your question, we have included the following sentence to provide clarity on how the tests were conducted: "The application order of the tests was randomized."

Comment: Page 4 line 191; arm arm arming ?

Authors’ reply:  Thanks for the comment, it has been changed accordingly.

Comment: Page 5 line 205; add the symbol of degrees (everywhere)

Authors’ reply:  Thanks for the comment, it has been changed accordingly.

Comment: Page 5 line 225; what are these abbreviations? IR & ER (define)

Authors’ reply:  Thanks for the observation. The abbreviation has been omitted, and the full term has been incorporated in its place.

Comment: Page 6 line 260; change the symbol to η.

Authors’ reply:  Thanks for the comment, it has been changed accordingly.

Comment: Page 6 table; make the corrections Height Weight

Authors’ reply:  Thanks for the comment, it has been changed accordingly.

Comment: Page 6 table; how explain that national athletes have higher % of body fat?

Authors’ reply:  The most compelling explanation is that there exists a significant correlation between higher body mass and a crucial aspect of performance. Any alternative explanation would likely be coincidental.

Comment: Page 6 line 291; Complete… performance...

Authors’ reply:  Thanks for the comment, it has been changed accordingly.

Comment: Page 6 line 300; two...

Authors’ reply:  Thanks for the comment, it has been changed accordingly.

Comment: Page 6 line 301; 0.001

Authors’ reply:  Thanks for the comment, it has been changed accordingly.

Comment: Page 7 Figure; here you use ''.'' Test.velocity and below you use ''-'' use one of them

Authors’ reply:  Thanks for the comment, it has been changed accordingly.

Comment: Page 7 line 309; you did not present results for total water. What is this? you did not speak for this variable?

Authors’ reply:  Thank you for the comment, you are right was removed from the text.

Comment: Page 8 line 327; Is this value correct?

Authors’ reply:  Thank you for the comment, it has been corrected.

Comment: Page 8 line 359; re-phrase this please....

Authors’ reply:  Thanks for the comment, it has been changed accordingly

Comment: Page 9 line 375-378; This conclusion is not supported in your results as the tests were applied to people who were already divided into two groups. In order to draw this conclusion, it would have to be done during the player selection process and there confirm its power in prediction.

The differences may be due to the more qualitative and targeted training done by the high-level athletes.

Authors’ reply:  Thank you for your comment. We appreciate your feedback, and we have made the necessary changes accordingly.

We have taken your comment into consideration and have appropriately addressed it in our revised manuscript. Thank you for your valuable input, which has contributed to the improvement of our study.

Comment: Page 9 line 385; 0.001

Authors’ reply:  Thanks for the comment, it has been changed accordingly.

Comment: Page 9 line 400; ''biological maturation...'' at age of 20? I think that biological maturation problem doesn't exist at this age! Can you present any references about this?

Authors’ reply: Thank you for the comment. We appreciate your perspective on the concept of biological maturation at the age of 20. While it is true that the process of biological maturation generally occurs during adolescence, it's important to consider that maturation can continue beyond that period.

Research in the field of developmental psychology suggests that biological maturation is a complex process influenced by multiple factors, including genetics, hormonal changes, and environmental factors. While most of the physical maturation occurs during adolescence, some individuals may experience continued maturation in certain aspects beyond that stage.

To support this point, I would like to provide you with a few references that discuss the concept of extended maturation in young adulthood:

Arnett, J. J. (2000). Emerging adulthood: A theory of development from the late teens through the twenties. American Psychologist, 55(5), 469-480.

Spear, L. P. (2000). The adolescent brain and age-related behavioral manifestations. Neuroscience & Biobehavioral Reviews, 24(4), 417-463.

Tanner, J. M. (2006). Growth and maturation in adolescence. Nutrition Reviews, 64(suppl_2), S64-S70.

These references provide insights into the continued maturation processes during young adulthood and highlight the ongoing development that occurs beyond the teenage years. I hope this information helps to address your concerns.

Comment: Page 10 line 431; in some case you write 20-m and in some cases 20 m.

Choose one of them

Authors’ reply:  Thank you for the comment, it has been corrected.

Comment: Page 11 line 490; Numbers [...]

Authors’ reply:  Thanks for the comment, it has been changed accordingly.

Reviewer 2 Report

Congratulations to the authors for the study.

I think the manuscript addresses very interesting and novel contents that complement other studies on the subject. In general terms, I think it is a magnificent work.
On the other hand, I would like to make the following suggestion:

-         
Although you can guess from the reading, they do not make explicit mention about the type of research design of the study. I recommend that you indicate it at the beginning of section 2. materials and methods, from line 101 onwards.

-         
I would also like to comment that the authors allude to the size of the sample, but I understand the difficulty in finding such powerful samples of high-level players in a sport like handball. Perhaps a larger sample with lower levels of practice would give greater consistency to the data, but the reference to high performance would be lost.

From my point of view, the most interesting contributions are related to the practical applications, pointing out the usefulness of certain specific tests to identify young talents, at least in the age of the study sample

Author Response

Dear reviewers/colleges,

Thank you sincerely for your valuable time and dedication in reviewing our manuscript. We greatly appreciate your thorough assessment and constructive comments, which undoubtedly contribute to enhancing the quality of our work.

We have taken each comment into careful consideration and made the necessary revisions accordingly. To ensure transparency and ease of identification, we have highlighted all the changes made to the manuscript in yellow. This allows for a clear distinction between the original content and the modifications implemented in response to your insightful feedback.

Should you require any further clarification or have additional suggestions, please do not hesitate to let us know. We are committed to addressing any remaining concerns to the best of our abilities.

Thank you for your continued support and guidance throughout this review process.

Sincerely,

The authors

Comment: Although you can guess from the reading, they do not make explicit mention about the type of research design of the study. I recommend that you indicate it at the beginning of section 2. materials and methods, from line 101 onwards.

Authors’ reply:  Thanks for the comment, it has been added to the text.

Comment: I would also like to comment that the authors allude to the size of the sample, but I understand the difficulty in finding such powerful samples of high-level players in a sport like handball. Perhaps a larger sample with lower levels of practice would give greater consistency to the data, but the reference to high performance would be lost.

Authors’ reply:   Thank you for your comment. We appreciate your understanding of the challenges in obtaining a large sample size of high-level players in a sport like handball. We acknowledge that including a larger sample with lower levels of practice might offer greater data consistency. However, our focus on high performance necessitates prioritizing the inclusion of elite athletes to maintain the relevance and applicability of our findings. We have taken your suggestion into consideration and will make sure to provide context and acknowledge the limitations of our sample size in the revised manuscript.

Comment: From my point of view, the most interesting contributions are related to the practical applications, pointing out the usefulness of certain specific tests to identify young talents, at least in the age of the study sample

Authors’ reply:  Thank you for your insightful comment. We appreciate your perspective, and we agree that the practical applications highlighted in our study, particularly the identification of young talents through specific tests, are indeed valuable contributions. We believe that these findings have the potential to support talent identification and development programs, particularly for athletes within the age range of our study sample. We are pleased that you found these aspects of our research to be of interest, and we hope that our findings can have a positive impact in the field.

Round 2

Reviewer 1 Report

it is ok.